# Altered reactivity to threatening stimuli in *Drosophila* models of Parkinson's disease, revealed by a trial-based assay

**Márton Kajtor[1], Viktor András Billes[2,3], Bálint Király[1,4,5], Patricia Karkusova[6], Tibor Kovács[2], Hannah Stabb[1], Katalin Sviatkó[1,7], Andor Vizi[1], Eszter Ujvári[1], Diána Balázsfi[1], Sophie E Seidenbecher[8], Duda Kvitsiani[9,10], Tibor Vellai[2,3], Balázs Hangya[1]***

[1]MTA–HUN-REN KOKI Lendület "Momentum" Laboratory of Systems Neuroscience, Institute of Experimental Medicine, Budapest, Hungary; [2]Department of Genetics, Eötvös Loránd University, Budapest, Hungary; [3]ELKH-ELTE Genetic Research Group, Budapest, Hungary; [4]Dynamics of Neural Systems Laboratory, AI Institute, Medical University of Vienna, Vienna, Austria; [5]Department of Biological Physics, Eötvös Loránd University, Budapest, Hungary; [6]Biomedical Center, Faculty of Medicine in Pilsen, Charles University, Pilsen, Czech Republic; [7]János Szentágothai Doctoral School of Neurosciences, Semmelweis University, Budapest, Hungary; [8]University of Technology Nuremberg, Nuremberg, Germany; [9]Department of Molecular Biology and Genetics, Danish Research Institute of Translational Neuroscience, Aarhus University, Aarhus, Denmark; [10]Department of Biomedical Sciences, Southern Illinois University, Carbondale, United States

*For correspondence:
hangya.balazs@koki.hun-ren.hu

Competing interest: The authors declare that no competing interests exist.

## eLife Assessment

The authors present **useful** findings on the use of a single-fly behavioral paradigm for assessing different *Drosophila* genetic models of neurodegeneration. The experimental design and analyses are **solid** and can be used for quick behavioral assessment in fly models of various neurodegenerative diseases, especially those having an impact on locomotion. The work will be of interest to *Drosophila* biologists using behavior as a readout for their studies.

**Abstract** The fruit fly *Drosophila melanogaster* emerges as an affordable, genetically tractable model of behavior and brain diseases. However, despite the surprising level of evolutionary conservation from flies to humans, significant genetic, circuit-level, and behavioral differences hinder the interpretability of fruit fly models for human disease. Therefore, to allow a more direct fly-versus-human comparison, we surveyed the rarely exploited, rich behavioral repertoire of fruit flies with genetic alterations relevant to Parkinson's disease (PD), including overexpression of human mutant Parkin or α-synuclein proteins and mutations in dopamine receptors. Flies with different genetic backgrounds displayed variable behaviors, including freezing, slowing, and running, in response to predator-mimicking passing shadows used as threatening stimuli in a single-animal trial-based assay. We found that the expression of human mutant Parkin in flies resulted in reduced walking speed and decreased reactivity to passing shadows. Flies with dopamine receptor mutations showed similar alterations, consistent with the motor and cognitive deficits typical in humans with PD. We also found age-dependent trends in behavioral choice during the fly lifespan, while dopamine receptor mutant flies maintained their decreased general reactivity throughout all age groups. Our data demonstrate that single-trial behavioral analysis can reveal subtle behavioral changes in mutant flies

that can be used to further our understanding of disease pathomechanisms and help gauge the validity of genetic *Drosophila* models of neurodegeneration, taking us one step closer to bridging the gap in fly-to-human translation.

## Introduction

Parkinson's disease (PD) affects over 6 million people worldwide, representing the second most common neurodegenerative disease. PD is slowly progressing and typically leads to years of aggravating disability, thereby placing a huge burden on families, health care systems, and society, measured in hundreds of billions of dollars annually (*Olesen et al., 2012*; *Obeso et al., 2017*; *Przedborski, 2017*; *Bloem et al., 2021*). Therefore, a hitherto elusive disease-modifying therapy is of prominent priority, envisioned to be fueled by research into basic mechanisms of the disease.

Patients with PD display a diverse set of motor and non-motor symptoms with an underlying progressive loss of midbrain dopaminergic neurons (*Schapira, 2009*; *Schapira et al., 2017*; *Bloem et al., 2021*), mechanisms of which are widely studied in animal models of the disease. Primate models offer the advantage of more direct translatability but are restricted to hard-to-handle neurotoxin approaches and do not promise fast progress. Unlike the genetic mouse models of Alzheimer's disease, most of the successful rodent models of PD are also toxin-based, making it difficult to exploit and further our understanding of the genetic bases of the disease (*Cannon and Greenamyre, 2010*; *Bové and Perier, 2012*; *Breger and Fuzzati Armentero, 2019*). This left a niche for the genetically tractable, affordable fruit flies as genetic models of PD (*Feany and Bender, 2000*; *Guo, 2012*; *Hewitt and Whitworth, 2017*), building on the homologies between the vertebrate basal ganglia and the fruit fly central complex (*Strausfeld and Hirth, 2013*). However, substantial genetic, anatomical, physiological, and behavioral discrepancies between insects and mammals call for better validation methods of fruit fly models for human disease. Therefore, to provide means for a more direct comparison between flies and mammals, we developed a single-animal trial-based behavioral assay that facilitates fine-grained assessment of phenotypical behavioral changes in fruit flies with genetic alterations relevant to understanding human PD.

Genes linked to familial forms of PD may serve as an ideal basis for genetic disease models. Notably, mutations in the *PARK2* gene, which encodes the Parkin protein involved in maintaining mitochondrial integrity, are associated with autosomal-recessive forms of PD (*Guo, 2012*). Parkin loss-of-function mutant flies were found to have advanced mitochondrial aging, structural mitochondrial damage, and a consequential selective loss of dopaminergic neurons (*Cackovic et al., 2018*), leading to motor deficits assessed by a climbing assay (*Chambers et al., 2013*; *Cackovic et al., 2018*), as well as non-motor PD phenotypes including memory deficits (*Julienne et al., 2017*). Mutations in the *SNCA* gene of α-Synuclein (α-Syn) are also associated with familial forms of PD (*Polymeropoulos et al., 1997*) and α-Syn has been shown to accumulate in Lewy bodies and neurites (*Spillantini et al., 1997*). α-Syn proteins have been found in the pre-synaptic terminals in humans and mice (*Kahle et al., 2000*) and are thought to be involved in regulating dopamine (DA) synthesis under physiological conditions by reducing tyrosine hydroxylase activity (*Perez et al., 2002*). Expression of human α-Syn in flies has been proposed as a genetic model of PD, showing age-dependent loss of dopaminergic neurons and locomotor dysfunction in a climbing assay (*Feany and Bender, 2000*; *Haywood and Staveley, 2006*). However, other studies found normal locomotion and dopaminergic cell counts in these flies, casting doubts on the validity of this model (*Pesah et al., 2005*; *Nagoshi, 2018*).

Despite the observed homology between mammalian and fruit fly DA systems in motor control and the establishment of *Drosophila* PD models based on human genetic information derived from familial PD patients, the role of *Drosophila* DA receptors in locomotor control is not well characterized. Nevertheless, it has been demonstrated that the D1-like DA receptor mediates ethanol-induced locomotion in the ellipsoid body (*Kong et al., 2010*). Furthermore, *Dop1R1* has been shown to be involved in turning behavior for goal-directed locomotion (*Kottler et al., 2019*) and startle-induced negative geotaxis (*Sun et al., 2018*). To shed light on the possible roles of DA receptors in threat-induced motor behaviors, we tested the behavioral responses of three dopamine receptor (*Dop1R1*, *Dop1R2*, and *DopEcR*) insertion mutant lines to predator-mimicking passing shadows and compared them to established fruit fly PD models with partially known locomotor deficits.

We found that flies expressing the R275W mutant allele of human Parkin ('Parkin flies') showed slower average locomotion speed, which, in contrast, was not a characteristic of flies expressing the A53T mutant allele of the human *SNCA* gene ('α-Syn flies'). Parkin flies also showed less behavioral reactivity to passing shadows compared to controls, whereas α-Syn flies showed increased durations of stopping after the stimuli. Dopamine receptor mutant flies showed reduced speed and less behavioral reactivity similar to Parkin flies. *Dop1R1* mutant flies exhibited more pronounced behavioral alterations than the other two receptor mutants in most of the parameters tested. These data demonstrate that mutations in DA receptor genes lead to specific patterns of behavioral deficits in *Drosophila*; hence, these dopamine receptor paralogs may have different functions in behavioral control. The modest phenotype of A53T α-Syn compared to Parkin flies suggests that the latter should be favored as a genetic model of human PD, at least with respect to the motor deficits examined in this study. We further propose that single-trial analyses such as those we present here help us gain a better understanding of the behavioral changes in fruit fly models of PD and are strong tools for validating *Drosophila* models of human diseases.

## Results

### A single-animal trial-based assay to test behavioral responses to predator-mimicking passing shadows

We designed a behavioral apparatus to examine the responses of individual flies to predator-mimicking passing shadows. To do this, we designed a transparent plexiglass arena (*Figure 1a*), featuring 13 tunnels (53 mm x 5 mm) for simultaneous tracking of 13 individual flies. The height of the tunnels (1.5 mm) was designed to allow free walking in two dimensions but prevent jumps and flight. To simulate predatory threat, we created passing 'shadows' (*Figure 1a*, right; see Methods) with a sliding red screen presented on a 10.1-inch display placed on top of the arena. A high frame-rate camera was placed under the arena (*Figure 1a*, left), allowing us to simultaneously record the movement of the animals and the shadows (*Figure 1—videos 1; 2*). All 13 tunnels housed a single fly in each session, enabling the collection of single-animal data from 13 flies in parallel. The locomotion of individual flies was tracked by custom software developed using the Bonsai visual programming environment (*Figure 1—figure supplement 1*, see Methods). We recorded 40-min-long sessions, consisting of 40 trials of 2-s-long shadow presentations separated by pseudorandom inter-trial intervals to prevent the animals from learning temporal expectations of the shadow presentations (*Figure 1b*).

Fruit flies exhibit a rich behavioral repertoire upon threatening stimuli, including freezing (or stopping) and various escape behaviors such as jumping, slow or fast take-off, and running, modulated by walking speed at the time of the threatening stimulus (*Zacarias et al., 2018*). To study these behaviors, we calculated the speed and acceleration of fruit flies based on the tracked x-y position of their center of mass (*Figure 2a*) and aligned these signals to the presentations of the shadow stimuli. To identify separate response types to the threatening stimuli, we applied hierarchical clustering on these stimulus-aligned speed traces (see Methods). This analysis consistently revealed three main stereotypical behavioral responses across flies in addition to the trials where no significant response was evoked (*Figure 2b*). Since jumping and flying were not possible in the arena, fruit flies were restricted to choose among freezing, slowing (also observed in *Zacarias et al., 2018*), and running. While this clustering approach was sufficient to reveal response types qualitatively, cluster boundaries were sensitive to genotype-specific differences across groups of flies (*Király and Hangya, 2022*). Therefore, for rigorous group comparisons, we determined exact definitions of each response type based on fly speed and acceleration (*Figure 2c*). Reactions were considered robust if the absolute value of the acceleration reached 200 mm/s$^2$; otherwise, the trial was classified as a 'no reaction' trial. We considered the reaction as a 'stop' if the animal was moving before shadow presentation and its speed decreased to zero in the first second relative to the shadow presentation. If fly speed decreased to a non-zero value, the trial was defined as 'slow down', and if the fly accelerated after the shadow presentation, the trial was classified as 'speed up' (see also Methods).

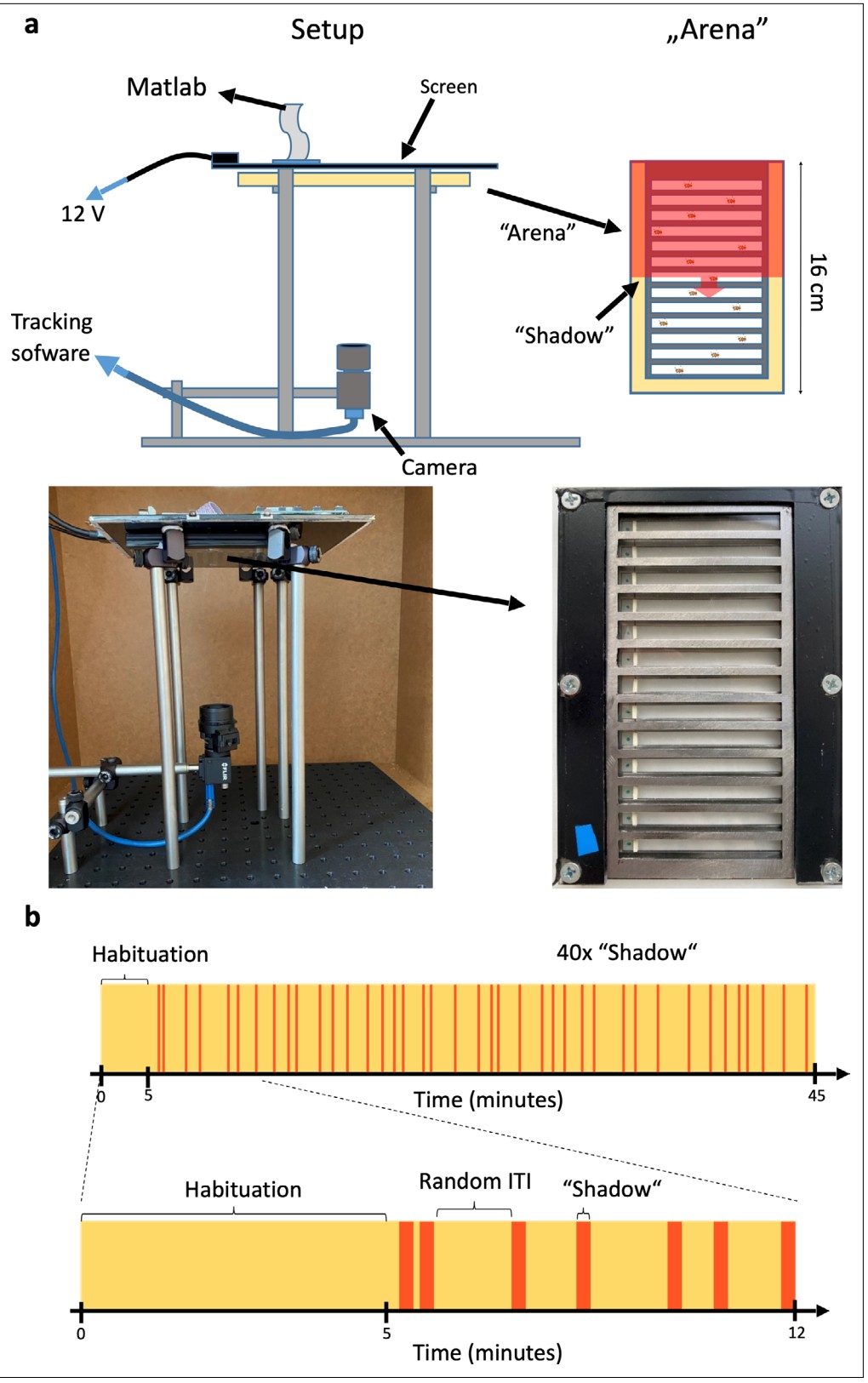

**Figure 1.** Behavioral apparatus and experimental design. (**a**) Top left, schematic of the experimental setup. Top right, schematic of the fly arena. Bottom left, photograph of the experimental setup. Bottom right, a photograph of the fly arena. (**b**) Timeline of a session showing the shadow presentations in red over the yellow background.

*Figure 1 continued on next page*

*Figure 1 continued*

The online version of this article includes the following video and figure supplement(s) for figure 1:

**Figure supplement 1.** Fly tracks from the camera view.

**Figure 1—video 1.** Example recording of *iso w$^{1118}$* files in the arena.

https://elifesciences.org/articles/90905/figures#fig1video1

**Figure 1—video 2.** Example recording of Parkin flies in the arena.

https://elifesciences.org/articles/90905/figures#fig1video2

## Parkinson's model and dopamine receptor mutant fruit flies showed reduced walking speed and decreased reactivity to threatening stimuli

To generate PD fly models, we expressed the human mutant Parkin (275 W) and α-Syn (A53T) coding transgenes (*UAS-Parkin-275W* and *UAS-α-Syn-A53T*, respectively), applying the UAS-Gal4 system (*Duffy, 2002*). Transgenes were driven by *Ddc-Gal4* inducing the expression of the correspondent human mutant proteins in dopaminergic and serotoninergic fly neurons. Parkin (275 W) and α-Syn (A53T) flies were compared to control animals from the same genetic background without mutant transgenes (*Ddc-Gal4* females were crossed with *isogenic w$^{1118}$* males; the examined F1 generation flies are referred to as *iso w$^{1118}$*; *Figure 3*) or mutants overexpressing GFP (*Figure 3—figure supplement 1*). The same level of eye pigmentation and vision of the compared genotypes was achieved by the prior replacement of the *w\** mutant X chromosome of the applied *Ddc-Gal4* stock for that of the wild-type. We also used Mi{MIC} random insertion lines for dopamine receptor mutants, namely *y$^1$ w\*; Mi{MIC}Dop1R1$^{MI04437}$* (BDSC 43773), *y$^1$ w\*; Mi{MIC}Dop1R2$^{MI08664}$* (BDSC 51098; *Pimentel et al., 2016*; *Harbison et al., 2019*), and *w$^{1118}$; PBac{PB}DopEcR$^{c02142}$/TM6B, Tb$^1$* (BDSC 10847; *Ishimoto et al., 2013*; *Petruccelli et al., 2016*; *Petruccelli et al., 2020*), referred to as Dop1R1, Dop1R2 and DopEcR, respectively. The dopamine receptor mutant groups were compared to their parental strains without the mutations (*y$^1$ w$^{67c23}$* served as control for *Dop1R1* and *Dop1R2* and *w$^{1118}$* for *DopEcR*; *Figure 3*).

To test whether mutant flies showed differences in overall locomotion independent of the threatening stimuli, we analyzed average fly speed in the 200ms time windows before stimulus presentation. We found that Parkin flies showed reduced mean speed compared to controls which expressed the Gal4 driver alone (24.96% reduction, p=6.55 × 10$^{-6}$, Mann-Whitney U-test; *Figure 3a*, top), while α-Syn flies did not show significant reduction (p=0.799, Mann–Whitney U-test). We observed similar changes in dopamine receptor mutant flies, where the *Dop1R1* and *DopEcR* defective lines showed a robust mean speed decrease compared to the control groups (*Dop1R1*, 19.68% decrease compared to *y$^1$ w$^{67c2}$*, p=0.0016; *DopEcR*, 32.97% decrease compared to *w$^{1118}$*, p=0.0034; *Figure 3a*, bottom), while the *Dop1R2* mutant flies only showed a non-significant speed decrease (21.73% compared to *y$^1$ w$^{67c23}$*, p=0.1009, Mann–Whitney U-test).

Next, we tested whether mutant flies showed a difference in their reaction to threatening stimuli. We found that PD model flies showed line-specific alterations in their freezing behavior. Parkin flies froze after stimuli less frequently, stopping in 37.72% of trials compared to the 43.48% observed in the *iso w$^{1118}$* animals (p=0.0043, Mann Whitney U-test; *Figure 3b*, top). However, in the stop trials, they showed normal duration of pauses in locomotion (p=0.8018 compared to *iso w$^{1118}$*, Mann Whitney U-test; *Figure 3c*, top). In contrast, α-Syn flies showed an unchanged stopping frequency when compared to controls (p=0.0768 compared to *iso w$^{1118}$*, Mann-Whitney U-test), but their stop durations showed a large increase (116.98% increase compared to *iso w$^{1118}$*, p=2.18 × 10$^{-11}$, Mann-Whitney U-test). Interestingly, in contrast to PD model flies, dopamine receptor mutant flies did not show significant differences in their freezing behavior relative to controls (stop proportion: *Dop1R1*, p=0.0631 compared to *y$^1$ w$^{67c23}$*; *Dop1R2*, p=0.1838 compared to *y$^1$ w$^{67c23}$*; *DopEcR*, p=0.1445 compared to *w$^{1118}$*; Mann-Whitney U-test; *Figure 3b*, bottom; stop duration: *Dop1R1*, p=0.893 compared to *y$^1$ w$^{67c23}$*; *Dop1R2*, p=0.154 compared to *y$^1$ w$^{67c23}$*; *DopEcR*, p=0.3576 compared to *w$^{1118}$*; Mann-Whitney U-test; *Figure 3c*, bottom).

α-Syn flies showed a reduced aptitude to increase their speed, or 'run', upon encountering threatening stimuli (26.67% decrease compared to *iso w$^{1118}$*, p=8.4 × 10$^{-6}$, Mann-Whitney U-test; *Figure 3d*, top). Similar results were found in *Dop1R1* (25% decrease compared to *y$^1$ w$^{67c23}$*, p=0.0122; Mann-Whitney U-test; *Figure 3d*, bottom) and *Dop1R2* (41.67% decrease compared to *y$^1$ w$^{67c23}$*, p=0.0024,

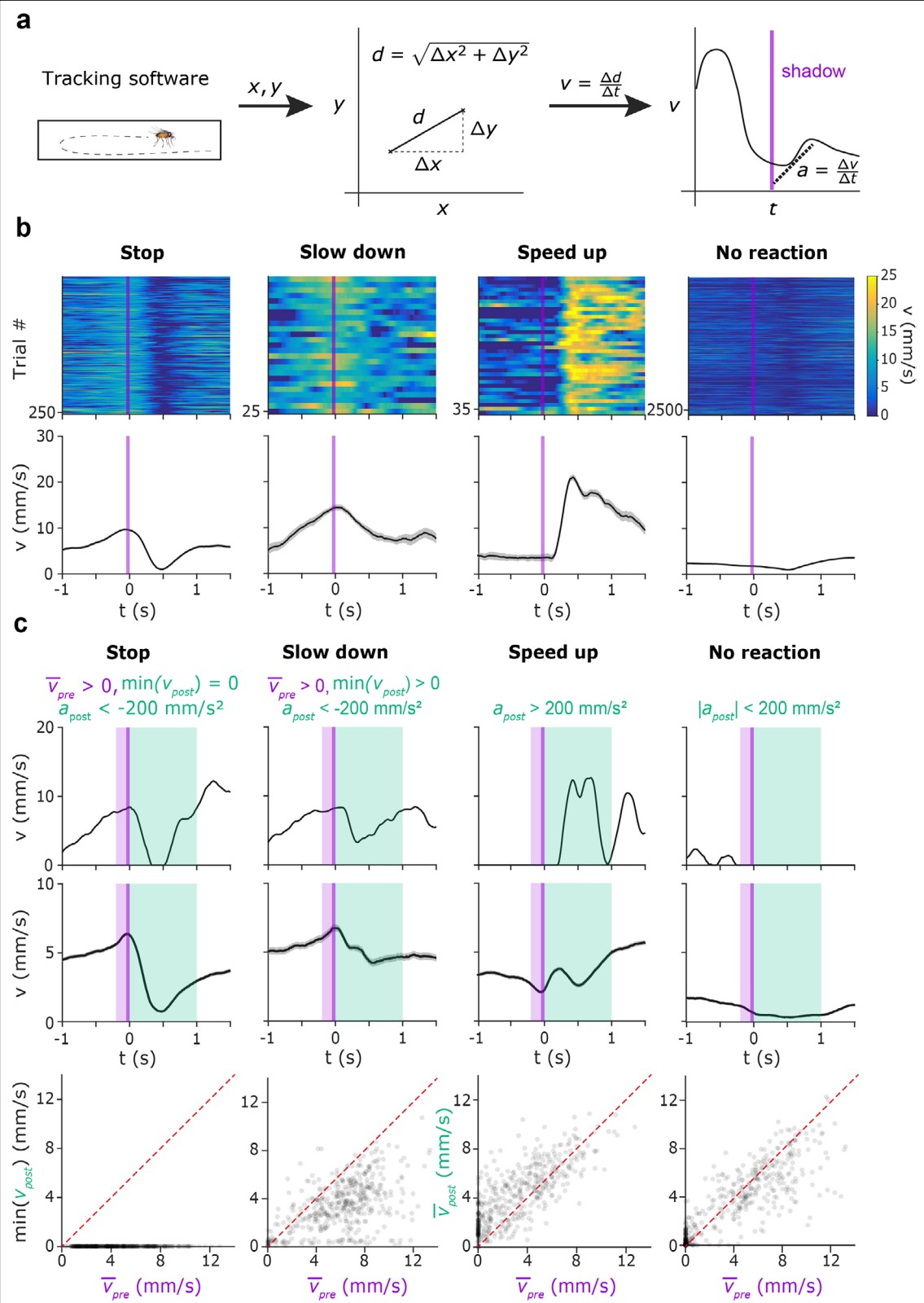

**Figure 2.** Behavioral characterization of the escape behavior repertoire of individual fruit flies. (**a**) Schematic for calculating speed and acceleration based on tracked position coordinates. (**b**) Four characteristic escape behaviors categorized by PCA for an example session of a control fly (w1118; from left; stop, slow down, speed up, no reaction). Top, color-coded heatmaps indicating the walking speed of the fly (blue, low speed; yellow, high speed), aligned to shadow presentations (purple line). Bottom, average moving speed triggered on the shadow presentations (purple line). Line and errorshade

*Figure 2 continued on next page*

Figure 2 continued

show mean ± SE. (**c**) Threshold-based classification of behavioral responses (from left, stop, slow down, speed up, no reaction). Top, single-trial example for each response type. Time intervals for calculating the average speed before the shadow presentation (purple) as well as the speed and acceleration after the shadow presentation (light green) are marked. Threshold values for each response type are displayed above the graphs. Middle, average walking speed across the trials from all sessions of $w^{1118}$ flies, sorted by the type of behavioral response. Line and errorshade show mean ± SE. Bottom, scatter plots showing minimal (Stop and Slow down trials) or average (Speed up and No reaction trials) post-stimulus speed vs. average pre-stimulus speed, representing the same trials as above.

Mann-Whitney U-test), but not in *DopEcR* mutant flies (compared to $w^{1118}$, p=0.6598, Mann-Whitney U-test). Frequency of slowing, that is reducing their speed without freezing in a full stop, was moderately decreased in Parkin (14.38% decrease compared to *iso* $w^{1118}$, p=0.0723, Mann-Whitney U-test; *Figure 3e*, top) and *DopEcR* mutant flies (15.28% decrease compared to *iso* $w^{1118}$, p=0.020, Mann-Whitney U-test; *Figure 3e*, bottom).

Overall, all mutations tested resulted in decreased reactivity to threatening stimuli, confirmed by a significant increase in the proportion of trials where no reactions were detected (*Figure 3f*). This effect was significant for Parkin flies (30% increase compared to *iso* $w^{1118}$, p=0.0043, Mann-Whitney U-test), as well as for *Dop1R1* (50% increase compared to $y^1 w^{67c23}$, p=0.0173, Mann-Whitney U-test) and *DopEcR* mutant flies (50% increase compared to *iso* $w^{1118}$, p=0.0167, Mann-Whitney U-test), but not for α-Syn (α-Syn vs. *iso* $w^{1118}$, p=0.151, Mann-Whitney U-test) and *Dop1R2* mutant flies (*Dop1R2* vs. $y^1 w^{67c23}$, p=0.1903, Mann-Whitney U-test).

While *Ddc-Gal4* lines are widely used (*Wang et al., 2007*), they express mutations both in dopaminergic and serotonergic neurons, preventing the assessment of whether the behavioral differences were due to one of those cell types alone. To address this, we tested *NP6510-Gal4-R275W* (Parkin) and *NP6510-Gal4-A53T* (α-Syn) as well as *TH-Gal4-R275W* (Parkin) and *TH-Gal4-A53T* (α-Syn) flies where expression of the mutation was restricted to dopaminergic cells only (*Riemensperger et al., 2013*). Results from these tests were largely similar to those of the *Ddc-Gal4* lines (*Figure 3— figure supplement 2*), reproducing the decreased speed (*NP6510-Gal4-R275W*, 30.3% decrease compared to *NP6510-Gal4- iso* $w^{1118}$, p=1.55 x $10^{-6}$; *NP6510-Gal4-A53T*, 20.4% decrease compared to *NP6510-Gal4- iso* $w^{1118}$, p=5.82 x $10^{-10}$; *TH-Gal4-R275W*, 13.08% decrease compared to *TH-Gal4- iso* $w^{1118}$, p=1.49 x $10^{-6}$; *TH-Gal4-A53T*, 17.57% decrease compared to *TH-Gal4- iso* $w^{1118}$, p=6.75 x $10^{-7}$; Mann–Whitney U-test) and decreased overall reactivity (*NP6510-Gal4-R275W*, 57.14% increase in 'No reaction' compared to *NP6510-Gal4- iso* $w^{1118}$, p=3.49$^{-8}$; *NP6510-Gal4-A53T*, 14.28% increase in 'No reaction' compared to *NP6510-Gal4- iso* $w^{1118}$, p=0.1658; *TH-Gal4-R275W*, 66.7% increase in 'No reaction' compared to *TH-Gal4- iso* $w^{1118}$, p=2.77 x $10^{-10}$; *TH-Gal4-A53T*, 66.7% increase in 'No reaction' compared to *TH-Gal4- iso* $w^{1118}$, p=1.11 x $10^{-6}$; Mann–Whitney U-test) of PD-model flies. Nevertheless, *TH-Gal4* and *NP6510-Gal4* mutants showed an increased propensity to stop (*NP6510-Gal4-R275W*, 16.47% increase compared to *NP6510-Gal4- iso* $w^{1118}$, p=4.4 x $10^{-11}$; *NP6510-Gal4-A53T*, 20.17% increase compared to *NP6510-Gal4- iso* $w^{1118}$, p=1.38 x $10^{-17}$; *TH-Gal4-R275W*, 16.32% increase compared to *TH-Gal4- iso* $w^{1118}$, p=1.55 x $10^{-4}$; *TH-Gal4-A53T*, 22.41% increase compared to *TH-Gal4- iso* $w^{1118}$, p=8.82 x $10^{-10}$, Mann–Whitney U-test). Stop duration showed a significant increase not only in α-Syn but also in Parkin fruit flies (*NP6510-Gal4-R275W*, 42.6% increase compared to *NP6510-Gal4- iso* $w^{1118}$, p=5.99$^{-5}$; *NP6510-Gal4-A53T*, 69.5% increase compared to *NP6510-Gal4- iso* $w^{1118}$, p=4.33 x $10^{-11}$; *TH-Gal4-R275W*, 33.62% increase compared to *TH-Gal4- iso* $w^{1118}$, p=1.63 x $10^{-6}$; *TH-Gal4-A53T*, 37.48% increase compared to *TH-Gal4- iso* $w^{1118}$, p=3.89 x $10^{-7}$; Mann–Whitney U-test). Overall, this suggests that mutations in the dopaminergic neurons alone were sufficient to produce the above-described phenotypical changes.

After these detailed comparisons of central tendencies of behavioral data, we also tested whether behavioral variability within fly response types showed differences across mutant groups. We found that stop duration showed a significantly larger dispersion in α-Syn flies compared to controls (p=0.0001, permutation test; *Figure 3—figure supplement 3*), while no other significant difference in data variance was detected.

## Reaction to threatening stimuli depends on fly walking speed

It has been shown that fruit flies may exhibit a different choice of escape behavior based on their momentary speed when encountering the threat (*Zacarias et al., 2018*). Therefore, we tested whether

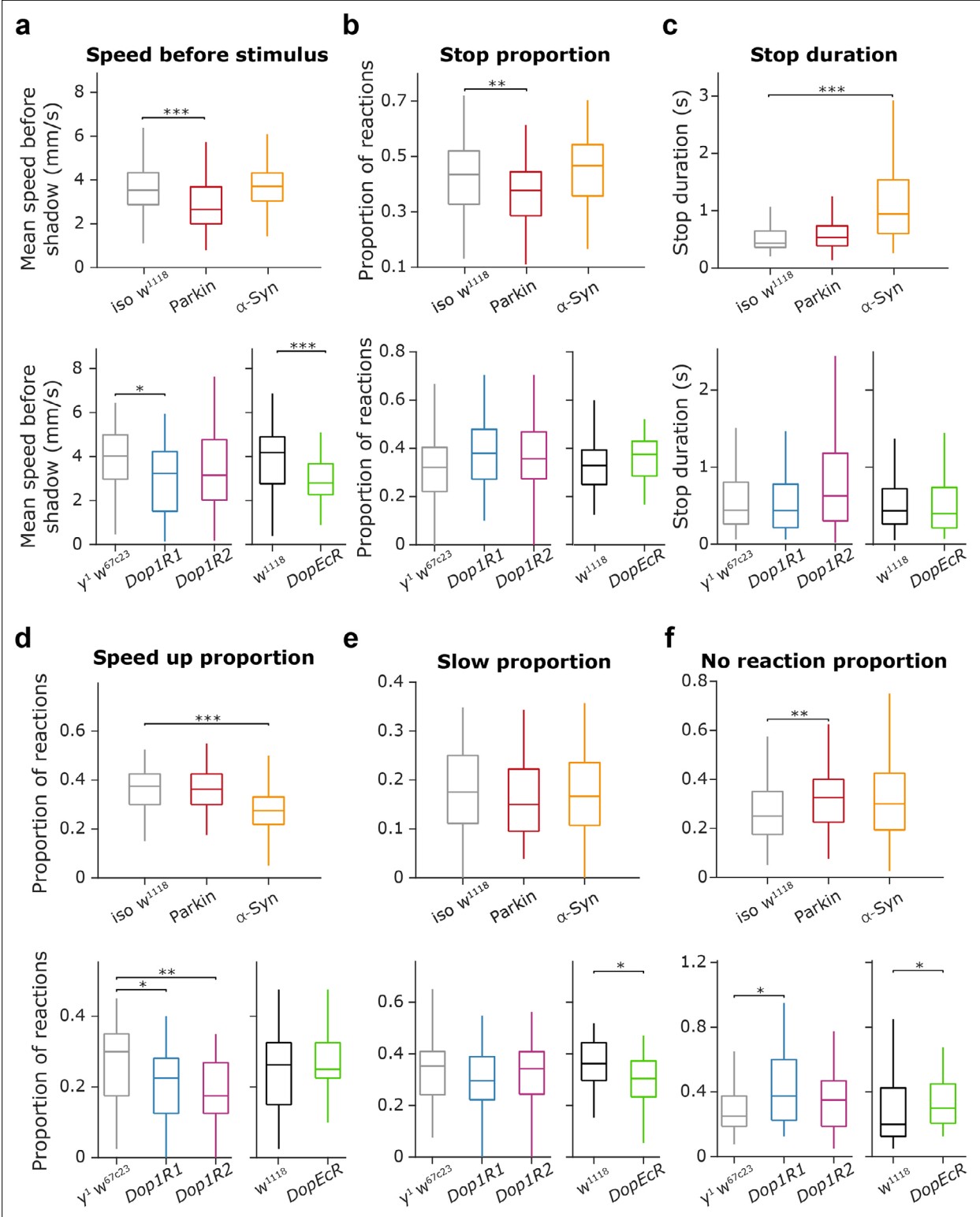

**Figure 3.** PD transgenic and dopamine receptor mutant fruit flies showed reduced walking speed and decreased reactivity to threatening stimuli.
(**a**) Distribution of the mean speed measured in the time window [–0.2, 0] seconds relative to shadow presentation for the different mutant groups.
Top, Parkin flies showed reduced mean speed compared to controls (Parkin vs. *iso w*$^{1118}$; p=6.55 × 10$^{-6}$, Mann–Whitney U-test). Bottom, *Dop1R1* and
*DopEcR* mutant flies showed reduced mean speed compared to controls (*y*$^{1}$*w*$^{67c23}$ and *w*$^{1118}$, respectively; p=0.0016, p=0.0034; Mann-Whitney U-test).
(**b**) Distribution of the proportion of stop trials in different mutant groups. Parkin flies showed a reduced tendency for stopping compared to *iso w*$^{1118}$
(p=0.0043, Mann Whitney U test). (**c**) Distribution of stop duration in different mutant groups. α-Syn flies showed increased stop durations compared to

*Figure 3 continued on next page*

*Figure 3 continued*

*iso w$^{1118}$* (p=2.18 × 10$^{-11}$, Mann-Whitney U-test). (**d**) Distribution of the proportion of speed-up trials in different mutant groups. Top, α-Syn flies showed a reduced tendency to speed up compared to their controls (p=8.4 × 10$^{-6}$, Mann-Whitney U-test). Bottom, *Dop1R1* and *Dop1R2* mutant flies also showed a significantly reduced tendency to speed up compared to their controls (p=0.0122 and p=0.0024, respectively; Mann-Whitney U-test). (**e**) Distribution of the proportion of slow down trials in different mutant groups. Bottom, *DopEcR* showed a 15.04% decrease compared to *w$^{1118}$* (p=0.020, Mann-Whitney U-test). (**f**) Distribution of the proportion of 'no reaction' trials in different mutant groups. Top, Parkin mutants showed reduced reactivity compared to *iso w$^{1118}$* controls (p=0.0043, Mann-Whitney U-test). Bottom, *Dop1R1* and *DopEcR* mutants also showed reduced reactivity compared to *y$^1$w$^{67c23}$* and *w$^{1118}$* controls, respectively (p=0.0173 and p=0.0167, respectively; Mann-Whitney U-test). Box-whisker plots show median, interquartile range, and non-outlier range. *, p<0.05; **, p<0.01; ***, p<0.001. Exact genotypes: *iso w$^{1118}$*: +; +; Ddc-Gal4/+. Parkin: +; +; Ddc-Gal4/UAS-Parkin-R275W. α-Syn: +; +; Ddc-Gal4/UAS-α-Syn-A53T. y$^1$ w$^{67c23}$: y$^1$ w$^{67c23}$. Dop1R: y$^1$ w*; Mi{MIC}Dop1R1$^{MI04437}$. Dop1R2: y$^1$ w*; Mi{MIC}Dop1R2$^{MI08664}$. DopEcR: w$^{1118}$; PBac{PB}DopEcR$^{c02142}$/ TM6B, Tb$^1$.

The online version of this article includes the following figure supplement(s) for figure 3:

**Figure supplement 1.** Escape behavior of Parkin and α-Syn flies compared to mutants overexpressing GFP.

**Figure supplement 2.** Restricting Parkin and α-Syn mutations to dopaminergic neurons.

**Figure supplement 3.** Stop duration showed large variance in α-Syn flies.

mutations in genes relevant to PD caused a change in this speed - behavioral response relationship. We calculated the probability of stopping, slowing, speeding up, and no reaction as a function of speed for all mutants, as well as the ratio of each response type conditioned on walking speed at the time of shadow presentations (*Figure 4a–b*). These analyses confirmed that 'running' and 'no reaction' were most likely at slow (<5 mm/s) or zero walking speeds, while slowing down was more frequent as the walking speed increased. Freezing was most frequently observed in the 5–13 mm/s speed range. We did not observe a significant difference between groups in their speed - behavioral response relationship for any of the response types (stops, p=0.4226; slow down, p=0.6025; speed up, p=0.9074; no reaction p=0.4692; two-way ANOVA genotype × speed interaction).

Since the distribution of the expression of various escape behaviors depended on walking speed, which was also different across the mutant lines tested (*Figure 3*), we asked whether the difference in baseline speed could explain the observed differences in behavioral reactions. Therefore, we performed simulations where reaction probabilities were based on the baseline walking speed distribution of each mutant line to test whether behavioral responses could be predicted based on speed alone (*Figure 4—figure supplement 1*). In most of the groups, we found a significant difference between the predicted and the measured response type distributions, suggesting a role for other behavioral differences among the mutant lines beyond the baseline speed differences (Parkin, p=0.0487; α-Syn, p<0.000001; *Dop1R1*, p=0.332; *Dop1R2*, p=0.006157; *DopEcR*, p=0.003901, chi-square test).

## Changes in escape behavior from the first to the fourth week of life

The age of *Drosophila* may have a significant influence on their responses to threatening stimuli. To test this, we examined control and dopamine receptor mutant flies (*w$^{1118}$, y$^1$ w$^{67c23}$, Dop1R1, Dop1R2, DopEcR*) in 5 age groups: from 1-day-old to 4-week-old (*Figure 5*; *Figure 5—figure supplement 1*). The animals were kept under conditions that accelerated their aging (29 °C and 70% humidity). Our data showed that the behavior of 1-day-old flies was significantly different from all other age groups. One-day-old flies stopped significantly more often, and the frequency of stopping gradually decreased with age in all groups (two-way ANOVA, age, p=1.87 × 10$^{-20}$, f=26.11; genotype, p=1.3 × 10$^{-29}$, f=38.42; genotype ×age, p=3.18 × 10$^{-6}$, f=3.54; 1-day-old vs. 1-week-old, p=1.1 × 10$^{-4}$; 1-week-old vs. 2-week-old, p=0.0074, 2-week-old vs. 3-week-old, p=0.1015; 3-week-old vs. 4-week-old, p=0.0079, Mann-Whitney test for post hoc analysis). In contrast, the probability of slowing down showed a substantial increase from 1-day-old to 1-week-old animals, and then remained relatively stable until the 4th week (two-way ANOVA, age, p=9.11 × 10$^{-15}$, f=18.72; genotype, p=1.4 × 10$^{-21}$, f=27.59; genotype ×age, p=0.0028, f=2.291; 1-day-old vs. 1-week-old, p=5.9 × 10$^{-10}$; 1-week-old vs. 2-week-old, p=0.992; 2-week-old vs. 3-week-old, p=0.0305; 3-week-old vs. 4-week-old, p=0.033; Mann-Whitney test for post hoc analysis). The probability of speeding up showed a similar trend, except that the 4-week-old animals showed a significant decrease compared to 3-week-old flies (two-way ANOVA, age, p=2.48 × 10$^{-15}$, f=19.44; genotype, p=2.28 × 10$^{-44}$, f=59.48; genotype ×age, p=1.1 × 10$^{-7}$, f=4.12; 1-day-old vs 1-week-old, p=2.64 × 10$^{-10}$; 1-week-old vs. 2-week-old, p=0.572; 2-week-old vs.

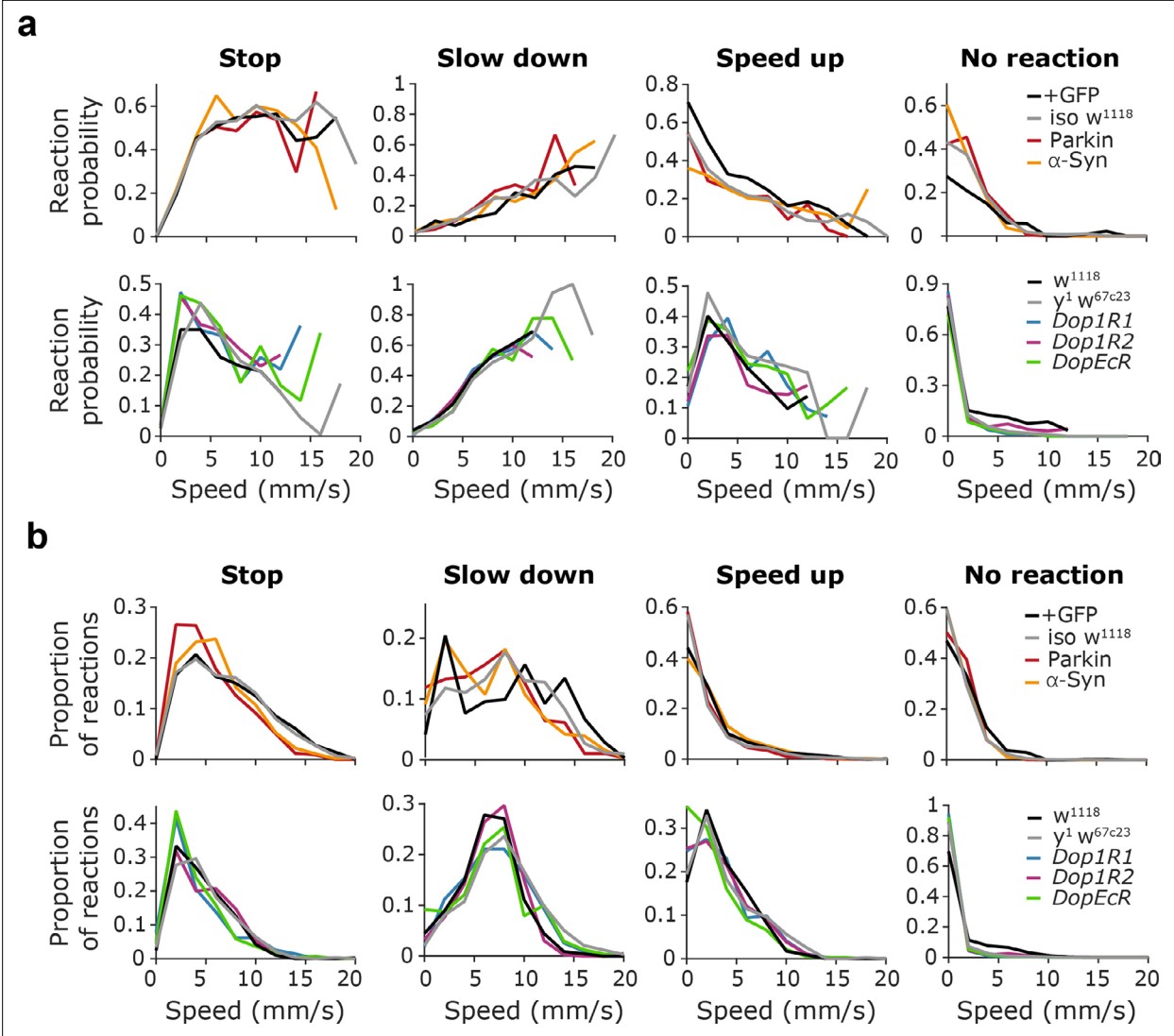

**Figure 4.** Reaction to threatening stimuli depends on fly walking speed. (**a**) Probability of a given response type as a function of average fly speed before the shadow presentation (200ms pre-stimulus time window). From left, stop, slow down, speed up, and no reaction trials are quantified. Top, Parkin, and α-Syn flies. Bottom, *Dop1R1*, *Dop1R2*, and *DopEcR* mutant flies. (**b**) Proportion of a given response type as a function of average fly speed before the shadow presentation. Top, Parkin and α-Syn flies. Bottom, *Dop1R1*, *Dop1R2*, and *DopEcR* mutant flies. Exact genotypes: *iso w^{1118}: +; +; Ddc-Gal4/+*. Parkin: *+; +; Ddc-Gal4/UAS-Parkin-R275W*. α-Syn: *+; +; Ddc-Gal4/UAS-α-Syn-A53T. y^1 w^{67c23}: y^1 w^{67c23}*. Dop1R: *y^1 w*; Mi{MIC}Dop1R1^{MI04437}*. *Dop1R2: y^1 w*; Mi{MIC}Dop1R2^{MI08664}*. DopEcR: *w^{1118}; PBac{PB}DopEcR^{c02142}/TM6B, Tb^1*.

The online version of this article includes the following figure supplement(s) for figure 4:

**Figure supplement 1.** Differences in walking speed do not explain different escape behaviors.

3-week-old, p=0.3445; 3-week-old vs. 4-week-old, p=0.0179; Mann-Whitney test for post hoc analysis). Thus, different types of escape reactions showed remarkable tendencies across the fly lifespan, while dopamine receptor mutant flies maintained their decreased general reactivity throughout all age groups.

## Discussion

We tested *Drosophila* mutant strains relevant for PD in a single-trial single-animal behavioral assay. Our tests revealed strain-specific behavioral alterations in flies' reactions to predator-mimicking passing shadows, serving as proof of principle demonstration of the viability of single-trial approaches in *Drosophila*, a method widely used in mammalian studies. Specifically, we found reduced walking

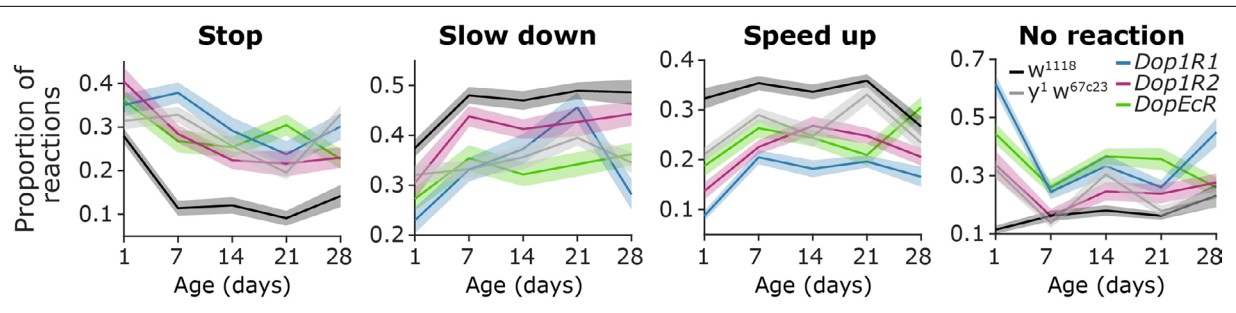

**Figure 5.** Changes in escape behavior from the first to the fourth week of life. Proportion of responses as a function of age for different groups of mutants and controls. From left to right, stop, slow down, speed up, and no reaction trials are quantified. Lines and error shades show mean and standard error. Exact genotypes: *iso w^{1118}: +; +; Ddc-Gal4/+*. Parkin: *+; +; Ddc-Gal4/UAS-Parkin-R275W*. α-Syn: *+; +; Ddc-Gal4/UAS-α-Syn-A53T. y^1 w^{67c23}: y^1 w^{67c23}*. Dop1R: *y^1 w*; Mi{MIC}Dop1R1^{MI04437}*. Dop1R2: *y^1 w*; Mi{MIC}Dop1R2^{MI08664}*. DopEcR: *w^{1118}; PBac{PB}DopEcR^{c02142}/TM6B, Tb^1*.

The online version of this article includes the following figure supplement(s) for figure 5:

**Figure supplement 1.** Changes in escape behavior from the first to the fourth week of life.

speed, decreased freezing frequency, and decreased overall reactivity in Parkin flies. In contrast, α-Syn flies merely showed an increased freezing duration without a concomitant change in freezing frequency, suggesting that Parkin flies better recapitulate some behavioral features of human PD progression. *Dop1R* mutant flies resembled Parkin flies in their decreased walking speed and decreased reactivity to the threatening stimuli. The distribution of the behavioral response fruit flies chose to execute depended on their speed at the time of stimuli, and this speed - behavioral response relationship was robust across the tested genotypes. Nevertheless, differences in walking speed alone did not explain the strain-specific behavioral alterations. Age dependence of behavioral choice was also found to be stereotypic across the tested genotypes.

Recent studies increasingly recognize the importance of investigating subtle changes in fly behavior to better understand the manifestations of locomotor and other disorders (*Geissmann et al., 2017*; *Zacarias et al., 2018*; *Seidenbecher et al., 2020*). Along these lines, Aggarwal and colleagues introduced an automated climbing assay for fruit flies and revealed subtle motor deficits in different mutant strains (*Aggarwal et al., 2019*). We expanded the scope of these studies by examining a diverse behavioral repertoire of *Drosophila* in response to threatening stimuli. Many earlier studies used looming stimulus, that is, a concentrically expanding shadow, mimicking the approach of a predator from above, to study escape responses in flies (*Card and Dickinson, 2008*; *de Vries and Clandinin, 2012*; *Zacarias et al., 2018*; *Ache et al., 2019*; *Oram and Card, 2022*) as well as rodents (*Lecca et al., 2017*; *Braine and Georges, 2023*; *Heinemans and Moita, 2024*). These assays have the advantage of closely resembling naturalistic, ecologically relevant threat-inducing stimuli and allow a relatively complete characterization of the fly escape behavior repertoire (*Zacarias et al., 2018*). As a flip side of their large degree of freedom, they do not lend themselves easily to provide a fully standardized, scalable behavioral assay. Therefore, Gibson et al. suggested a novel threat-inducing assay operating with moving overhead translational stimuli, that is, passing shadows, and demonstrated that they induce escape behaviors in flies akin to looming discs (*Gibson et al., 2015*). This assay, coined ReVSA (repetitive visual stimulus-induced arousal) by the authors, had the advantage of scalability, while constraining flies to a walking arena that somewhat restricted the remarkably rich escape types flies otherwise exhibit. Here we carried this idea one step further by using a screen to present the shadows instead of a physically moving paddle and putting individual flies to linear corridors instead of the common circular fly arena. This ensured that the shadow reached the same coordinates in all linear tracks concurrently and made it easy to accurately determine when individual flies encountered the stimulus, aiding data analysis and scalability. We found the same escape behavioral repertoire as in studies with looming stimuli and ReVSA (*Gibson et al., 2015*; *Zacarias et al., 2018*), with a similar dependence on walking speed (*Zacarias et al., 2018*; *Oram and Card, 2022*), confirming that looming stimuli and passing shadows can both be considered as threat-inducing visual stimuli.

Mutations in the *PARK2* gene lead to impaired ubiquitination and a consequential mitochondrial dysfunction (*Guo, 2010*; *Guo, 2012*), and early-onset PD in human patients. This led to the widespread use of Parkin flies as a genetic model of PD (*Guo, 2010*; *Chambers et al., 2013*; *Aggarwal*

*et al., 2019*). Of note, flies that lack Parkin display a significant degeneration of dopaminergic neurons (*Whitworth et al., 2005*). We found similarities between Parkin and *Dop1R* mutant flies in their altered responses to predator-mimicking passing shadows: reduced walking speed and decreased overall reactivity, suggesting that the lack of dopamine action through *Dop1R* may be one of the common pathways underlying motor deficits. This is in line with a recent study demonstrating impaired startle-induced geotaxis and locomotor reactivity in *Dop1R* mutant flies (*Sun et al., 2018*), similar to what had been shown for Parkin flies (*Aggarwal et al., 2019*), and a demonstration of dopaminergic control over walking speed (*Marquis and Wilson, 2022*). In contrast, flies expressing human α-syn showed moderate changes in motor behavior except for a marked prolongation of freezing duration upon threatening stimuli. This is in accordance with studies suggesting that α-syn misexpression is not fully penetrant under some conditions (*Pesah et al., 2005*; *Nagoshi, 2018*), and calls for the use of other genetic *Drosophila* models, including Parkin flies, in studying the pathophysics of human PD.

The dopamine/ecdysteroid receptor *DopEcR* is a G-protein-coupled dual receptor for dopamine and the steroid hormone ecdysone. It has been proposed that the *DopEcR* may serve as an integrative hub for dopamine-mediated actions and stress responses in fruit flies (*Petruccelli et al., 2020*). We found that *DopEcR* mutant flies showed decreased mean walking speed, decreased probability of slowing down, and decreased overall behavioral reactivity in response to predator-mimicking passing shadows. This pattern of alterations was largely similar to those observed in Parkin and *Dop1R* mutant flies, suggesting that the *DopEcR* may convey similar dopamine-mediated functions as *DopR1,* at least in those motor aspects tested in the present study. A recent work demonstrated that serotonin also modulates both walking speed and startle response in flies, suggesting a complex neuromodulatory control over *Drosophila* motor behavior (*Howard et al., 2019*). We recognize a limitation in using PBac{PB}DopEcRc02142 over the TM6B, Tb1 balancer chromosome, as the balancer itself may induce behavioral deficits in flies. We consider this unlikely, as the PBac{PB}DopEcRc02142 mutation demonstrates behavioral effects even in heterozygotes (*Ishimoto et al., 2013*). Additionally, to our knowledge, no studies have reported behavioral deficits in flies carrying the TM6B, Tb1 balancer chromosome over a wild-type chromosome.

In humans, degeneration of midbrain dopaminergic neurons has been found as a direct cause of PD symptoms, and DA supplementation with its precursor levodopa has been one of the most successful therapeutic approaches to date. The fruit fly and mammalian DA systems show a number of homologies which are thought to serve as a good basis for *Drosophila* PD models (*Feany and Bender, 2000*; *Hewitt and Whitworth, 2017*). Specifically, a deep-running evolutional conservation has been revealed between the arthropod central complex and the vertebrate basal ganglia, where GABAergic and dopaminergic neurons play key roles in motor control in both phyla (*Strausfeld and Hirth, 2013*). Clusters of fly dopaminergic neurons that project to the ellipsoid body, the fan-shaped body, and the lateral accessory lobes are thought to share homologies with the striatum-projecting dopaminergic neurons of the substantia nigra in mammals. Both fruit fly central complex and vertebrate basal ganglia mediate a range of functions from motor control and sensorimotor integration to action selection and decision making to motivation (*Strauss, 2002*; *Claridge-Chang et al., 2009*; *Kong et al., 2010*) and learning (*McCurdy et al., 2021*; *Zolin et al., 2021*; *Fisher et al., 2022*; *Qiao et al., 2022*; *Taisz and Jefferis, 2022*; *Yamada et al., 2023*). It has been found that fruit flies lacking the DA-synthesizing tyrosine hydroxylase enzyme in the central nervous system show reduced activity and locomotor deficit that worsen with age and also exhibit marked impairments in associative learning based on both appetitive and aversive reinforcement (*Riemensperger et al., 2011*). This is in line with mammalian studies emphasizing the role of the dopaminergic nigrostriatal pathway in controlling goal-directed action (*Eban-Rothschild et al., 2016*), reinforcement learning (*Schultz et al., 1997*; *Lak et al., 2014*), and more recently, conveying aversive information (*Menegas et al., 2015*; *Menegas et al., 2018*). Our findings are consistent with the apparent homologies across fruit fly and mammalian dopaminergic systems and suggest that a more in-depth investigation of specific behavioral changes and underlying dopamine-related dysfunctions may yield translatable results. We also observed age-related changes in motor behavior in response to threatening stimuli, particularly in 4-week-old flies, which may parallel the age-dependent worsening of PD symptoms in humans.

**Table 1.** Number of recorded flies for age group comparisons.

|  | *Dop1R1* | *Dop1R2* | *DopEcR* | *w[1118]* | *y[1] w[67c23]* |
|---|---|---|---|---|---|
| 1-day-old | 35 | 36 | 36 | 35 | 41 |
| 1-week-old | 36 | 54 | 36 | 36 | 36 |
| 2-week-old | 36 | 36 | 36 | 36 | 30 |
| 3-week-old | 35 | 35 | 29 | 36 | 36 |
| 4-week-old | 30 | 36 | 34 | 30 | 30 |

## Methods

### Animals

We used 2-week-old *Drosophila melanogaster* (both males and females) raised at 24 °C and 60% humidity in a natural light-dark cycle for experiments presented in *Figures 1–4*. For age-group comparisons (*Figure 5*), fruit flies aged 1, 7, 14, 21, and 28 days were used. These flies were raised at 29 °C and 70% humidity to accelerate aging.

Three sets of mutant groups were used for behavioral comparison. In the first set of animals, we used *w\*; UAS-Parkin-R275W* (created in the laboratory of Kah-Leong Lim, Neurodegeneration Research Laboratory, National Neuroscience Institute, Singapore; *Wang et al., 2007*; *Kovács et al., 2017*) and *w\*; UAS-alpha-Syn-A53T* (BDSC 8148) from the Eotvos Lorand University (*Szinyákovics et al., 2023*), with *y[1], v[1]; UAS-GFP* (BDSC 35786) and *+; P{Ddc-GAL4.L}4.36* (modified version of BDSC 7009, the original first chromosome *w[1118]* has been replaced to w[+] in this study) as controls. For the second set of animals, we used *Dop1R1*, *Dop1R2*, and *DopEcR* mutant flies kindly donated by the Anne Von Phillipsborn lab, DANDRITE, Aarhus University and *y[1] w\*; Mi{MIC}Dop1R1[MI04437]* (BDSC 43773), *y[1] w\*; Mi{MIC}Dop1R2[MI08664]* (BDSC 51098), and *w[1118]; PBac{PB}DopEcR[c02142]/TM6B, Tb[1]* (BDSC 10847) with *w[1118]* (BDSC 5905) *y[1]* and *y[1] w[67c23]* (BDSC 6599) as controls from the Bloomington stock center. *Table 1* shows the number of recorded flies for the age group comparisons, and *Table 2* shows the number of recorded flies for comparing the mutant groups. For the third set of flies, we used w[\*]; P{w[+mC]=ple-GAL4.F}3 (*TH-Gal4*), Bloomington *Drosophila* Stock Center: 8848 (*Friggi-Grelin et al., 2003*), and y[\*] w[\*]; P{w[+mW.hs]=GawB}NP6510 (*NP6510-Gal4*), Kyoto DGGR: 113956

**Table 2.** Number of recorded flies for comparing the mutant groups.

| 2-week-old | No. of flies |
|---|---|
| *Dop1R1* | 45 |
| *Dop1R2* | 47 |
| *DopEcR* | 35 |
| *w[1118]* | 46 |
| *y[1], w[67c23]* | 40 |
| Parkin (Ddc) | 78 |
| α-Syn (Ddc) | 69 |
| *iso w[1118]* | 92 |
| +GFP | 65 |
| Parkin (NP6510) | 170 |
| α-Syn (NP6510)_ | 181 |
| +/ NP6510-Gal4 | 176 |
| Parkin (TH) | 322 |
| α-Syn (TH) | 176 |
| +/ TH-Gal4 | 282 |

(*Riemensperger et al., 2013*; *Majcin Dorcikova et al., 2023*). Since it was important to ensure that the animals' light perception was not impaired, we replaced the white loss-of-function mutant allele of the transgenic *Gal4* lines with a wild-type allele. As controls, we used transheterozygous +/ *TH-Gal4* and +/ *NP6510-Gal4* strains in all cases.

### Arena

We designed an arena made of four layers of plexiglass and metal that formed 13 tunnels (*Figure 1a*). Each tunnel was 52 mm x 5 mm ×1.5 mm, in which individual flies were able to move freely in two dimensions but could not fly. The bottom layer was a standard transparent plexiglass layer. The second layer contained the tunnel walls made of metal to prevent horizontal spread of light. The third layer was the top of the tunnels, made of transparent plexiglass, which could slide above the tunnel. It also contained a small hole on the side, where the flies could be inserted into the tunnel through a pipette. The fourth layer was a metal cover designed with cut-outs that matched the shape and position of the tunnels to prevent flies in the neighboring tunnels from detecting the shadow before it reached their position.

### Movement tracking

Flies were tracked by using a FLIR Camera (FLIR Blackfly S USB3 FLIR Systems, Wilsonville, OR, US) placed under the arena (*Figure 1a*). The frame rate was set to 100 frames/second to allow the detection of position with high temporal resolution. The recorded images were processed by Bonsai (2.5.2) tracking software (*Lopes et al., 2015*), using customized Bonsai code. The code extracted each tunnel area separately from the image and detected flies by selecting the biggest and darkest pixel object in the tunnel area in real time. Fly position coordinates were calculated from the centroid. The software stored x-y position coordinates along with their timestamps in csv files, which were later processed in Matlab R2016a (Mathworks, Natick, MA, US).

### Predator-mimicking shadow stimuli

A 10.1-inch screen (HannStar HSD101PWW1-A00) was placed on top of the arena. The screen presented a yellow background. We implemented 'shadow' stimuli in red color, as these stimuli were suggested to be perceived by fruit flies as dark shadows and also enabled continuous tracking of the animals (*Sharkey et al., 2020*). The red color screen represented a large enough contrast change to evoke escape behaviors of flies but also provided sufficient brightness to enable continuous motion tracking. A passing shadow stimulus was chosen, as it was affecting all tunnels simultaneously. To mimic an approaching shadow, a red screen slid in from the side, stayed on for 2 s, then slid out. The shadow stimuli were separated by pseudorandomized inter-trial intervals with an approximately exponential distribution with a mean of 52.2 s, preventing flies from anticipating the next shadow stimulus.

### Identification of behavioral response types to the threatening stimuli

Animal speed was calculated as a function of time in a –200 ms to 1000 ms window around each shadow stimulus in 10ms time bins. These speed functions were normalized by subtracting the average speed in the 200ms window before the shadow presentation to reveal stimulus-induced absolute instantaneous speed changes. Principal component analysis (PCA) was used to reduce dimensionality of the normalized speed functions in the 1 s interval following shadow presentation in a way that the variance between trials is maximally preserved in the low-dimensional representation. Agglomerative hierarchical clustering was performed in the space spanned by the first three principal components to identify trials with distinct characteristic response types.

### Behavior detection

After identifying the four most frequently observed animal responses to shadow presentations, we algorithmically defined them based on the speed and acceleration thresholds. First, we examined the speed of the animal in the 200ms time window before the shadow ($v_{pre}$); stopping/freezing or slowing down was only possible if the fly had been moving, defined by $v_{pre}$ >0 mm/s. Second, we examined the acceleration ($a_{post}$) in the 1 s interval following the shadow using a 100ms moving average window to characterize the flies' response. Reactions were considered robust if the absolute value of the acceleration reached 200 mm/s$^2$. A trial was characterized as a 'stop' trial if the average $v_{pre}$ was above

0 mm/s, the fly's first reaction was deceleration, and 0 mm/s speed was reached before accelerating again. If the first reaction was deceleration but 0 mm/s speed was not reached, the trial was classified as a 'slow down' trial. In 'speed up' trials, the first response following the shadow was acceleration. If the absolute value of $a_{post}$ did not reach the 200 mm/s$^2$ threshold during the 1 s interval following the shadow, the trial was labeled as a 'no reaction' trial.

## Data analysis and statistics

Data analysis was performed in Matlab R2016a (Mathworks, Natick, MA, US) using custom-written code. Statistical comparison between groups of flies was performed by two-sided Mann-Whitney U-test. Exact p values are reported in the Results section. We performed a multi-group comparison in the 'Changes in escape behavior from the first to the fourth week of life' section using two-way ANOVA, as it is reportedly robust to non-normality of the data (*Mooi et al., 2018*; *Knief and Forstmeier, 2021*), while there is a lack of a consensual non-parametric alternative. Nevertheless, Kruskal-Wallis tests for the main effect of age showed significant effects in all genotypes in accordance with the ANOVA, confirming the results (Stop frequency, *DopEcR* p=0.0007; *Dop1R1*, p=0.004; *Dop1R2*, p=9.94 × 10$^{-5}$; *w$^{1118}$*, p=9.89 × 10$^{-13}$; *y$^1$ w$^{67c23}$*, p=2.54 × 10$^{-5}$; Slowing down frequency, *DopEcR*, p=0.0421; *Dop1R1*, p=5.77 x 10$^{-6}$; *Dop1R2*, p=0.011; *w$^{1118}$*, p=2.62 x 10$^{-5}$; *y$^1$ w$^{67c23}$*, p=0.0382; Speeding up frequency, *DopEcR*, p=0.0003; *Dop1R1*, p=2.06 x 10$^{-7}$; *Dop1R2*, p=2.19 x 10$^{-6}$; *w$^{1118}$*, p=0.0044; *y$^1$ w$^{67c23}$*, p=1.36 x 10$^{-5}$). We used the Mann-Whitney U-test for post hoc analyses to maintain consistent use of tests across figures. Since there is no straightforward way of correcting for multiple comparisons in this case, we reported uncorrected p values and suggest considering them at p=0.01, which minimizes type I errors. Nevertheless, Tukey's tests with the 'honest significance' approach yielded similar results. The effect of movement speed on the distribution of behavioral response types was tested using a nested Monte Carlo simulation framework (*Figure 4—figure supplement 1*). This simulation aimed to model how different movement speeds impact the probability distribution of response types, comparing these simulated outcomes to empirical data. This approach allowed us to determine whether observed differences in response distributions were solely due to speed variations across genotypes or if additional behavioral factors contributed to the differences. First, we calculated the probability of each response type at different specific speed values (outer model). These probabilities were derived from the grand average of all trials across each genotype, capturing the overall tendency at various speeds. Second, we simulated the behavior of virtual flies (n=3,000 per genotypes, which falls within the same order of magnitude as the number of experimentally recorded trials from different genotypes) by drawing random velocity values from the empirical velocity distribution specific to the given genotype and then randomly selecting a reaction based on the reaction probabilities associated with the drawn velocity (inner model). Finally, we calculated reaction probabilities for the virtual flies and compared them with real data from animals of the same genotype. Differences were statistically tested by Chi-squared test.

## Acknowledgements

We thank the Anne Von Phillipsborn lab, DANDRITE, Aarhus University for the kind gift of the DA receptor mutant lines. Most of the *Drosophila* stocks were obtained from the Bloomington *Drosophila* Stock Center (NIH P40OD018537). We thank the FENS-Kavli Network of Excellence for fruitful discussions. BH was supported by the Eotvos Lorand Research Network, the Hungarian Research Network, the Hungarian Brain Research Program NAP3.0 (NAP2022-I-1/2022) and NAP-KOLL-2023-1/2023 grant by the Hungarian Academy of Sciences, the "Lendület" LP2024-8/2024 grant by the Hungarian Academy of Sciences, and the NKFIH K147097 grant of the National Research, Development and Innovation Office. TV was supported by the grants OTKA (Hungarian Scientific Research Fund) K132439 and VEKOP (VEKOP-2.3.2-16-2017-00014), and by the ELKH/MTA-ELTE Genetics Research Group (01062). TK was supported by the OTKA grant (Hungarian Scientific Research Fund) PD 143786, by the University Excellence Fund of Eötvös Loránd University, Budapest, Hungary (EKA_2022/045-P302-1), and by the National Research Excellence Programme STARTING 150612.

# Additional information

## Funding

| Funder | Grant reference number | Author |
| --- | --- | --- |
| Hungarian Academy of Sciences | NAP2022-I-1/2022 | Balázs Hangya |
| Hungarian Academy of Sciences | NAP-KOLL-2023-1/2023 | Balázs Hangya |
| Hungarian Academy of Sciences | LP2024-8/2024 | Balázs Hangya |
| National Research, Development and Innovation Office | K147097 | Balázs Hangya |
| National Research, Development and Innovation Office | K132439 | Tibor Vellai |
| National Research, Development and Innovation Office | VEKOP-2.3.2-16-2017-00014 | Tibor Vellai |
| National Research, Development and Innovation Office | PD143786 | Tibor Kovács |
| Eötvös Loránd University | EKA_2022/045-P302-1 | Tibor Kovács |
| National Research, Development and Innovation Office | STARTING 150612 | Tibor Kovács |
| ELKH/MTA-ELTE Genetics Research Group | 01062 | Tibor Vellai |
| Eotvos Lorand Research Network | | Balázs Hangya |
| Hungarian Research Network | | Balázs Hangya |

The funders had no role in study design, data collection and interpretation, or the decision to submit the work for publication.

## Author contributions

Márton Kajtor, Data curation, Software, Formal analysis, Investigation, Visualization, Methodology, Writing – review and editing; Viktor András Billes, Tibor Kovács, Resources, Methodology, Writing – review and editing; Bálint Király, Software, Formal analysis, Validation, Investigation, Visualization, Methodology, Writing – review and editing; Patricia Karkusova, Data curation, Formal analysis, Investigation, Writing – review and editing; Hannah Stabb, Andor Vizi, Eszter Ujvári, Data curation, Writing – review and editing; Katalin Sviatkó, Diána Balázsfi, Supervision, Writing – review and editing; Sophie E Seidenbecher, Methodology, Writing – review and editing; Duda Kvitsiani, Conceptualization, Resources, Supervision, Methodology, Writing – review and editing; Tibor Vellai, Resources, Supervision, Methodology, Writing – review and editing; Balázs Hangya, Conceptualization, Supervision, Funding acquisition, Writing – original draft, Writing – review and editing

## Author ORCIDs

Bálint Király ⓘ https://orcid.org/0000-0001-8483-8780
Balázs Hangya ⓘ https://orcid.org/0000-0003-2709-7407

Reviewer #1 (Public review): https://doi.org/10.7554/eLife.90905.3.sa1
Reviewer #2 (Public review): https://doi.org/10.7554/eLife.90905.3.sa2
Author response https://doi.org/10.7554/eLife.90905.3.sa3

## Additional files

### Supplementary files
MDAR checklist

### Data availability
All analysis functions are available at GitHub (copy archived at *Hangya, 2025*). All data are available at figshare.

The following dataset was generated:

| Author(s) | Year | Dataset title | Dataset URL | Database and Identifier |
|---|---|---|---|---|
| Kajtor M, Király B, Hangya B | 2025 | Altered reactivity to threatening stimuli in *Drosophila* models of Parkinson's disease revealed by a trial-based assay | https://doi.org/10.6084/m9.figshare.23659626 | figshare, 10.6084/m9.figshare.23659626 |

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
