## [Editor Report · eLife Assessment]

The authors present **useful** findings on the use of a single-fly behavioral paradigm for assessing different *Drosophila* genetic models of neurodegeneration. The experimental design and analyses are **solid** and can be used for quick behavioral assessment in fly models of various neurodegenerative diseases, especially those having an impact on locomotion. The work will be of interest to *Drosophila* biologists using behavior as a readout for their studies.

---

## [Referee Report · Reviewer #1 (Public review)]

Translating discoveries from model organisms to humans is often challenging, especially in neuropsychiatric diseases, due to the vast gaps in the circuit complexities and cognitive capabilities. Kajtor et al. propose to bridge this gap in the fly models of Parkinson's disease (PD) by developing a new behavioural assay where flies respond to a moving shadow by modifying their locomotor activities. The authors believe the flies' response to the shadow approximates their escape response to an approaching predator. To validate this argument, they tested several PD-relevant transgenic fly lines and showed that some of them indeed have altered responses in their assay.

Strengths:

This single-fly-based assay is easy and inexpensive to set up, scalable and provides sensitive, quantitative estimates to probe flies' optomotor acuity. The behavioural data is detailed, and the analysis parameters are well-explained.

Weaknesses:

The authors have yet to link cellular physiology to behaviour. It will be interesting to see how future use of this assay helps uncover connections between cellular pathology and behavioural changes.

---

## [Referee Report · Reviewer #2 (Public review)]

The manifestation and progression of neurodegenerative disorders is poorly understood. Many of the neuronal disorders start by presenting subtle changes in neuronal circuit and quantification and measurement of these subtle behavior responses could help one delineate the mechanisms involved. The present study very nicely uses the flies' behavioral response to predator-mimicking passing shadows to measure subtle changes in their behavior. The data from various fly genetic models of Parkinson's disease supports their claim. This single trial method is useful to capture the individual animal's response to the threatening stimuli but stops short of capturing the fine ambulatory responses which could provide further information on an individual's behavioral response. By capturing the fine features, the authors could get detailed observations, such as posture, gait or wing positioning for a better understanding the behavioral response to the passing shadow.

---

## [Author Response]

The following is the authors’ response to the original reviews

We thank the Reviewers for their constructive comments and the Editor for the possibility to address the Reviewers’ points in this rebuttal. We

(1) Conducted new experiments with *NP6510-Gal4* and *TH-Gal4* lines to address potential behavioral differences due to targeting dopaminergic vs. both dopaminergic and serotonergic neurons

(2) Conducted novel data analyses to emphasize the strength of sampling distributions of behavioral parameters across trials and individual flies

(3) Provided Supplementary Movies

(4) Calculated additional statistics

(5) Edited and added text to address all points of the Reviewers.

Please see our point-by-point responses below.

**Public Reviews:**

**Reviewer #1 (Public Review):**
Summary:Translating discoveries from model organisms to humans is often challenging, especially in neuropsychiatric diseases, due to the vast gaps in the circuit complexities and cognitive capabilities. Kajtor et al. propose to bridge this gap in the fly models of Parkinson's disease (PD) by developing a new behavioral assay where flies respond to a moving shadow by modifying their locomotor activities. The authors believe the flies' response to the shadow approximates their escape response to an approaching predator. To validate this argument, they tested several PD-relevant transgenic fly lines and showed that some of them indeed have altered responses in their assay.Strengths:This single-fly-based assay is easy and inexpensive to set up, scalable, and provides sensitive, quantitative estimates to probe flies' optomotor acuity. The behavioral data is detailed, and the analysis parameters are well-explained.

We thank the Reviewer for the positive assessment of our study.

Weaknesses:While the abstract promises to give us an assay to accelerate fly-to-human translation, the authors need to provide evidence to show that this is indeed the case. They have used PD lines extensively characterized by other groups, often with cheaper and easier-to-setup assays like negative geotaxis, and do not offer any new insights into them. The conceptual leap from a low-level behavioral phenotype, e.g. changes in walking speed, to recapitulating human PD progression is enormous, and the paper does not make any attempt to bridge it. It needs to be clarified how this assay provides a new understanding of the fly PD models, as the authors do not explore the cellular/circuit basis of the phenotypes. Similarly, they have assumed that the behavior they are looking at is an escape-from-predator response modulated by the central complex- is there any evidence to support these assumptions? Because of their rather superficial approach, the paper does not go beyond providing us with a collection of interesting but preliminary observations.

We thank the Reviewer for pointing out some limitations of our study. We would like to emphasize that what we perceive as the main advantage of performing single-fly and single-trial analyses is the access to rich data distributions that provide more fine-scale information compared to bulk assays. We think that this is exactly going one step closer to ‘bridging the enormous conceptual leap from a low-level behavioral phenotype, e.g. changes in walking speed, to recapitulating human PD progression’, and we showcase this in our study by comparing the distributions over the entire repertoire of behavioral responses across fly mutants. Nevertheless, we agree with the Reviewer that many more steps in this direction are needed to improve translatability. Therefore, we toned down the corresponding statements in the Abstract and in the Introduction. Moreover, to further emphasize the strength of sampling distributions of behavioral parameters across trials and individual flies, we complemented our comparisons of central tendencies with testing for potential differences in data dispersion, demonstrated in the novel Supplementary Figure S4.

Looming stimuli have been used to characterize flies’ escape behaviors. These studies uncovered a surprisingly rich behavioral repertoire (Zacarias et al., 2018), which was modulated by both sensory and motor context, e.g. walking speed at time of stimulus presentation (Card and Dickinson, 2008; Oram and Card, 2022; Zacarias et al., 2018). The neural basis of these behaviors was also investigated, revealing loom-sensitive neurons in the optic lobe and the giant fiber escape pathway (Ache et al., 2019; de Vries and Clandinin, 2012). Although less frequently, passing shadows were also employed as threat-inducing stimuli in flies (Gibson et al., 2015). We opted for this variant of the stimulus so that we could ensure that the shadow reached the same coordinates in all linear track concurrently, aiding data analysis and scalability. Similar to the cited study, we found the same behavioral repertoire as in studies with looming stimuli, with an equivalent dependence on walking speed, confirming that looming stimuli and passing shadows can both be considered as threat-inducing visual stimuli. We added a discussion on this topic to the main text.

**Reviewer #2 (Public Review):**
In this study, Kajtor et al investigated the use of a single-animal trial-based behavioral assay for the assessment of subtle changes in the locomotor behavior of different genetic models of Parkinson's disease of *Drosophila*. Different genotypes used in this study were Ddc-GAL4>UASParkin-275W and UAS- α-Syn-A53T. The authors measured *Drosophila's* response to predatormimicking passing shadow as a threatening stimulus. Along with these, various dopamine (DA) receptor mutants, Dop1R1, Dop1R2 and DopEcR were also tested.The behavior was measured in a custom-designed apparatus that allows simultaneous testing of 13 individual flies in a plexiglass arena. The inter-trial intervals were randomized for 40 trials within 40 minutes duration and fly responses were defined into freezing, slowing down, and running by hierarchical clustering. Most of the mutant flies showed decreased reactivity to threatening stimuli, but the speed-response behavior was genotype invariant.These data nicely show that measuring responses to the predator-mimicking passing shadows could be used to assess the subtle differences in the locomotion parameters in various genetic models of *Drosophila*.The understanding of the manifestation of various neuronal disorders is a topic of active research. Many of the neuronal disorders start by presenting subtle changes in neuronal circuits and quantification and measurement of these subtle behavior responses could help one delineate the mechanisms involved. The data from the present study nicely uses the behavioral response to predator-mimicking passing shadows to measure subtle changes in behavior. However, there are a few important points that would help establish the robustness of this study.

We thank the Reviewer for the constructive comments and the positive assessment of our study.

(1) The visual threat stimulus for measuring response behavior in *Drosophila* is previously established for both single and multiple flies in an arena. A comparative analysis of data and the pros and cons of the previously established techniques (for example, Gibson et al., 2015) with the technique presented in this study would be important to establish the current assay as an important advancement.

We thank the Reviewer for this suggestion. We included the following discussion on measuring response behavior to visual threat stimuli in the revised manuscript.

Many earlier studies used looming stimulus, that is, a concentrically expanding shadow, mimicking the approach of a predator from above, to study escape responses in flies (Ache et al., 2019; Card and Dickinson, 2008; de Vries and Clandinin, 2012; Oram and Card, 2022; Zacarias et al., 2018) as well as rodents (Braine and Georges, 2023; Heinemans and Moita, 2024; Lecca et al., 2017). These assays have the advantage of closely resembling naturalistic, ecologically relevant threatinducing stimuli, and allow a relatively complete characterization of the fly escape behavior repertoire. As a flip side of their large degree of freedom, they do not lend themselves easily to provide a fully standardized, scalable behavioral assay. Therefore, Gibson et al. suggested a novel threat-inducing assay operating with moving overhead translational stimuli, that is, passing shadows, and demonstrated that they induce escape behaviors in flies akin to looming discs (Gibson et al., 2015). This assay, coined ReVSA (repetitive visual stimulus-induced arousal) by the authors, had the advantage of scalability, while constraining flies to a walking arena that somewhat restricted the remarkably rich escape types flies otherwise exhibit. Here we carried this idea one step further by using a screen to present the shadows instead of a physically moving paddle and putting individual flies to linear corridors instead of the common circular fly arena. This ensured that the shadow reached the same coordinates in all linear tracks concurrently and made it easy to accurately determine when individual flies encountered the stimulus, aiding data analysis and scalability. We found the same escape behavioral repertoire as in studies with looming stimuli and ReVSA (Gibson et al., 2015; Zacarias et al., 2018), with a similar dependence on walking speed (Oram and Card, 2022; Zacarias et al., 2018), confirming that looming stimuli and passing shadows can both be considered as threat-inducing visual stimuli.

(2) Parkinson's disease mutants should be validated with other GAL-4 drivers along with DdcGAL4, such as NP6510-Gal4 (Riemensperger et al., 2013). This would be important to delineate the behavioral differences due to dopaminergic neurons and serotonergic neurons and establish the Parkinson's disease phenotype robustly.

We thank the Reviewer for point out this limitation. To address this, we repeated our key experiments in Fig.3. with both TH-Gal4 and NP6510-Gal4 lines, and their respective controls. These yielded largely similar results to the Ddc-Gal4 lines reported in Fig.3., reproducing the decreased speed and decreased overall reactivity of PD-model flies. Nevertheless, TH-Gal4 and NP6510-Gal4 mutants showed an increased propensity to stop. Stop duration showed a significant increase not only in α-Syn but also in Parkin fruit flies. These novel results have been added to the text and are demonstrated in Supplementary Figure S3.

(3) The DopEcR mutant genotype used for behavior analysis is w1118; PBac{PB}DopEcRc02142TM6B, Tb1. Balancer chromosomes, such as TM6B,Tb can have undesirable and uncharacterised behavioral effects. This could be addressed by removing the balancer and testing the DopEcR mutant in homozygous (if viable) or heterozygous conditions.

We appreciate the Reviewer's comment and acknowledge the potential for the *DopEcR* balancer chromosome to produce unintended behavioral effects. However, given that this mutant was not essential to our main conclusions, we opted not to repeat the experiment. Nevertheless, we now discuss the possible confounds associated with using the PBac{PB}DopEcRc02142 mutant allele over the balancer chromosome. “We recognize a limitation in using PBac{PB}DopEcRc02142 over the TM6B, *Tb1* balancer chromosome, as the balancer itself may induce behavioral deficits in flies. We consider this unlikely, as the PBac{PB}DopEcRc02142 mutation demonstrates behavioral effects even in heterozygotes (Ishimoto et al., 2013). Additionally, to our knowledge, no studies have reported behavioral deficits in flies carrying the TM6B, *Tb1* balancer chromosome over a wild-type chromosome.”

(4) The height of the arena is restricted to 1mm. However, for the wild-type flies (Canton-S) and many other mutants, the height is usually more than 1mm. Also, a 1 mm height could restrict the fly movement. For example, it might not allow the flies to flip upside down in the arena easily. This could introduce some unwanted behavioral changes. A simple experiment with an arena of height at least 2.5mm could be used to verify the effect of 1mm height.

We thank the Reviewer for this comment, which prompted us to reassess the dimensions of the apparatus. The height of the arena was 1.5 mm, which we corrected now in the text. We observed that the arena did not restrict the flies walking and that flies could flip in the arena. We now include two Supplementary Movies to demonstrate this.

(5) The detailed model for Monte Carlo simulation for speed-response simulation is not described. The simulation model and its hyperparameters need to be described in more depth and with proper justification.

We thank the Reviewer for pointing out a lack of details with respect to Monte Carlo simulations. We used a nested model built from actual data distributions, without any assumptions. Accordingly, the stimulation did not have hyperparameters typical in machine learning applications, the only external parameter being the number of resamplings (3000 for each draw). We made these modeling choices clearer and expanded this part as follows.

“The effect of movement speed on the distribution of behavioral response types was tested using a nested Monte Carlo simulation framework (Fig. S5). This simulation aimed to model how different movement speeds impact the probability distribution of response types, comparing these simulated outcomes to empirical data. This approach allowed us to determine whether observed differences in response distributions are solely due to speed variations across genotypes or if additional behavioral factors contribute to the differences. First, we calculated the probability of each response type at different specific speed values (outer model). These probabilities were derived from the grand average of all trials across each genotype, capturing the overall tendency at various speeds. Second, we simulated behavior of virtual flies (n = 3000 per genotypes, which falls within the same order of magnitude as the number of experimentally recorded trials from different genotypes) by drawing random velocity values from the empirical velocity distribution specific to the given genotype and then randomly selecting a reaction based on the reaction probabilities associated with the drawn velocity (inner model). Finally, we calculated reaction probabilities for the virtual flies and compared it with real data from animals of the same genotype.

Differences were statistically tested by Chi-squared test.”

(6) The statistical analysis in different experiments needs revisiting. It wasn't clear to me if the authors checked if the data is normally distributed. A simple remedy to this would be to check the normality of data using the Shapiro-Wilk test or Kolmogorov-Smirnov test. Based on the normality check, data should be further analyzed using either parametric or non-parametric statistical tests. Further, the statistical test for the age-dependent behavior response needs revisiting as well. Using two-way ANOVA is not justified given the complexity of the experimental design. Again, after checking for the normality of data, a more rigorous statistical test, such as split-plot ANOVA or a generalized linear model could be used.

We thank the Reviewer for this comment. We performed Kolmogorov-Smirnov test for normality on the data distributions underlying Figure 3, and normality was rejected for all data distributions at p = 0.05, which justifies the use of the non-parametric Mann-Whitney U-test. Regarding ANOVA, we would like to point out that the ANOVA hypothesis test design is robust to deviations from normality (Knief and Forstmeier, 2021; Mooi et al., 2018). While the Kruskal-Wallis test is considered a reasonable non-parametric alternative of one-way ANOVA, there is no clear consensus for a non-parametric alternative of two-way ANOVA. Therefore, we left the two-way ANOVA for Figure 5 in place; however, to increase the statistical confidence in our conclusions, we performed Kruskal-Wallis tests for the main effect of age and found significant effects in all genotypes in accordance with the ANOVA, confirming the results (Stop frequency, *DopEcR* p = 0.0007; *Dop1R1*, p = 0.004; *Dop1R2*, p = 9.94 × 10^-5^; *w1118*, p = 9.89 × 10^-13^; *y1 w67c23*, p = 2.54 × 10^-5^; Slowing down frequency, *DopEcR*, p = 0.0421; *Dop1R1*, p = 5.77 x 10^-6^; *Dop1R2*, p = 0.011; *w1118*, p = 2.62 x 10^-5^; *y1 w67c23*, p = 0.0382; Speeding up frequency, *DopEcR*, p = 0.0003; *Dop1R1*, p = 2.06 x 10^-7^; *Dop1R2*, p = 2.19 x 10^-6^; *w1118*, p = 0.0044; *y1 w67c23*, p = 1.36 x 10^-5^). We also changed the post hoc Tukey-tests to post hoc Mann-Whitney tests in the text to be consistent with the statistical analyses for Figure 3. These resulted in very similar results as the Tukey-tests. Of note, there isn’t a straightforward way of correcting for multiple comparisons in this case as opposed to the Tukey’s ‘honest significance’ approach, we thus report uncorrected p values and suggest considering them at p = 0.01, which minimizes type I errors. These notes have been added to the ‘Data analysis and statistics’ Methods section.

(7) The dopamine receptor mutants used in this study are well characterized for learning and memory deficits. In the Parkinson's disease model of *Drosophila*, there is a loss of DA neurons in specific pockets in the central brain. Hence, it would be apt to use whole animal DA receptor mutants as general DA mutants rather than the Parkinson's disease model. The authors may want to rework the title to reflect the same.

We thank the Reviewer for this comment, which suggests that we were not sufficiently clear on the *Drosophila* lines with DA receptor mutations. We used Mi{MIC} random insertion lines for dopamine receptor mutants, namely *y1 w*1; Mi{MIC}Dop1R1MI04437* (BDSC 43773), *y1 w*1; Mi{MIC}Dop1R2MI08664* (BDSC 51098) (Harbison et al., 2019; Pimentel et al., 2016), and *w1118; PBac{PB}DopEcRc02142/TM6B, Tb1* (BDSC 10847) (Ishimoto et al., 2013; Petruccelli et al., 2020, 2016). These lines carried reported mutations in dopamine receptors, most likely generating partial knock down of the respective receptors. We made this clearer by including the full names at the first occurrence of the lines in Results (beyond those in Methods) and adding references to each of the lines.

**Recommendations for the authors:**

**Reviewer #1 (Recommendations For The Authors):**
(1) Please think about focusing the manuscript either on the escape response or the PD pathology and provide additional evidence to demonstrate that you indeed have a novel system to address open questions in the field.

As detailed above, we now emphasize more that the main advantage of our single-trial-based approach lies in the appropriate statistical comparison of rich distributions of behavioral data. Please see our response to the ‘Weaknesses’ section for more details.

(2) Please explain the rationale for choosing the genetic lines and provide appropriate genetic controls in the experiments, e.g. trans-heterozygotes. Why use Ddc-Gal4 instead of TH or other specific Split-Gal4 lines?

We thank the Reviewer for this suggestion. We repeated our key experiments with *TH-Gal4* and *NP6510-Gal4* lines. Please see our response to Point #2 of Reviewer #2 for details.

(3) Please proofread the manuscript for ommissions. e.g. there's no legend for Fig 4b.

We respectfully point out that the legend is there, and it reads “b, Proportion of a given response type as a function of average fly speed before the shadow presentation. Top, Parkin and α-Syn flies. Bottom, *Dop1R1*, *Dop1R2* and *DopEcR* mutant flies.”

**Reviewer #2 (Recommendations For The Authors):**
(1) In figure 2(c), representing the average walking speed data for different mutants would be useful to visually correlate the walking differences.

We thank the Reviewer for this suggestion. The average walking speed was added in a scatter plot format, as suggested in the next point of the Reviewer.

(2) The data could be represented more clearly using scatter plots. Also, the color scheme could be more color-blindness friendly.

We thank the Reviewer for this suggestion. We added scatter plots to Fig.2c that indeed represent the distribution of behavioral responses better. We also changed the color scheme and removed red/green labeling.

(3) The manuscript should be checked for typos such as in line 252, 449, 484.

Thank you. We fixed the typos.

References

Ache JM, Polsky J, Alghailani S, Parekh R, Breads P, Peek MY, Bock DD, von Reyn CR, Card GM. 2019. Neural Basis for Looming Size and Velocity Encoding in the *Drosophila* Giant Fiber Escape Pathway. Curr Biol 29:1073-1081.e4. doi:10.1016/j.cub.2019.01.079

Braine A, Georges F. 2023. Emotion in action: When emotions meet motor circuits. Neurosci Biobehav Rev 155:105475. doi:10.1016/j.neubiorev.2023.105475

Card G, Dickinson MH. 2008. Visually Mediated Motor Planning in the Escape Response of *Drosophila*. Curr Biol 18:1300–1307. doi:10.1016/j.cub.2008.07.094

de Vries SEJ, Clandinin TR. 2012. Loom-Sensitive Neurons Link Computation to Action in the *Drosophila* Visual System. Curr Biol 22:353–362. doi:10.1016/j.cub.2012.01.007

Gibson WT, Gonzalez CR, Fernandez C, Ramasamy L, Tabachnik T, Du RR, Felsen PD, Maire MR, Perona P, Anderson DJ. 2015. Behavioral Responses to a Repetitive Visual Threat Stimulus Express a Persistent State of Defensive Arousal in *Drosophila*. Curr Biol 25:1401– 1415. doi:10.1016/j.cub.2015.03.058

Harbison ST, Kumar S, Huang W, McCoy LJ, Smith KR, Mackay TFC. 2019. Genome-Wide Association Study of Circadian Behavior in *Drosophila melanogaster*. Behav Genet 49:60–82. doi:10.1007/s10519-018-9932-0

Heinemans M, Moita MA. 2024. Looming stimuli reliably drive innate defensive responses in male rats, but not learned defensive responses. Sci Rep 14:21578. doi:10.1038/s41598-02470256-2

Ishimoto H, Wang Z, Rao Y, Wu C, Kitamoto T. 2013. A Novel Role for Ecdysone in *Drosophila* Conditioned Behavior: Linking GPCR-Mediated Non-canonical Steroid Action to cAMP Signaling in the Adult Brain. PLoS Genet 9:e1003843. doi:10.1371/journal.pgen.1003843

Knief U, Forstmeier W. 2021. Violating the normality assumption may be the lesser of two evils. Behav Res Methods 53:2576–2590. doi:10.3758/s13428-021-01587-5

Lecca S, Meye FJ, Trusel M, Tchenio A, Harris J, Schwarz MK, Burdakov D, Georges F, Mameli M. 2017. Aversive stimuli drive hypothalamus-to-habenula excitation to promote escape behavior. Elife 6:1–16. doi:10.7554/eLife.30697

Mooi E, Sarstedt M, Mooi-Reci I. 2018. Market Research, Springer Texts in Business and Economics. Singapore: Springer Singapore. doi:10.1007/978-981-10-5218-7

Oram TB, Card GM. 2022. Context-dependent control of behavior in *Drosophila*. Curr Opin Neurobiol 73:102523. doi:10.1016/j.conb.2022.02.003

Petruccelli E, Lark A, Mrkvicka JA, Kitamoto T. 2020. Significance of DopEcR, a G-protein coupled dopamine/ecdysteroid receptor, in physiological and behavioral response to stressors. J Neurogenet 34:55–68. doi:10.1080/01677063.2019.1710144

Petruccelli E, Li Q, Rao Y, Kitamoto T. 2016. The Unique Dopamine/Ecdysteroid Receptor Modulates Ethanol-Induced Sedation in *Drosophila*. J Neurosci 36:4647–4657. doi:10.1523/JNEUROSCI.3774-15.2016

Pimentel D, Donlea JM, Talbot CB, Song SM, Thurston AJF, Miesenböck G. 2016. Operation of a homeostatic sleep switch. Nature 536:333–337. doi:10.1038/nature19055

Zacarias R, Namiki S, Card GM, Vasconcelos ML, Moita MA. 2018. Speed dependent descending control of freezing behavior in *Drosophila melanogaster*. Nat Commun 9:1–11. doi:10.1038/s41467-018-05875-1